# Multifunctional electrolyte additive for high power lithium metal batteries at ultra-low temperatures

Weili Zhang[1,2,5] ✉, Yang Lu [1,5], Qingqing Feng[2,5], Hao Wang[2], Guangyu Cheng[3], Hao Liu[2], Qingbin Cao[4], Zhenjun luo[2], Pan Zhou[1], Yingchun Xia[1], Wenhui Hou[1], Kun Zhao[2], Chunyi Du[2] & Kai Liu [1] ✉

Ultra-low-temperature lithium metal batteries face significant challenges, including sluggish ion transport and uncontrolled lithium dendrite formation, particularly at high power. An ideal electrolyte requires high carrier ion concentration, low viscosity, rapid de-solvation, and stable interfaces, but balancing these attributes remains a formidable task. Here, we design and synthesize a multifunctional additive, perfluoroalkylsulfonyl quaternary ammonium nitrate ($PQA\text{-}NO_3$), which features both cationic ($PQA^+$) and anionic ($NO_3^-$) components. $PQA^+$ reacts in situ with lithium metal to form an inorganic-rich solid-electrolyte interphase (SEI) that enhances $Li^+$ transport through the SEI film. $NO_3^-$ creates an anion-rich, solvent-poor solvation structure, improving oxidation stability at the positive electrode/electrolyte interface and reducing $Li^+$-solvent interactions. This allows ether-based electrolytes to achieve high voltage tolerance, increased ionic conductivity, and lower de-solvation energy barriers. The Li (40 μm)∥ NMC811 (3 mAh cm$^{-2}$) coin cells with the developed electrolyte exhibited stable cycling at -60 °C and a 450 Wh kg$^{-1}$ pouch cell retained 48.1% capacity at -85 °C, achieving a specific energy (except tabs and packing foil, same hereafter) of 171.8 Wh kg$^{-1}$. Additionally, the pouch cell demonstrated a discharge rate of 3.0 C at -50 °C, reaching a specific power (except tabs and packing foil, same hereafter) of 938.5 W kg$^{-1}$, indicating the electrolyte's suitability for high-rate lithium metal batteries in extreme low-temperature environments.

The development of lithium metal batteries (LMBs) has garnered significant attention due to their potential to deliver high-specific energy. However, the thermodynamic instability of lithium metal and the significant volume changes during deposition and stripping processes lead to the fragility of the solid-electrolyte interphase (SEI) on the lithium metal surface. This instability results in the growth of dendritic lithium and the formation of dead lithium during repeated deposition/ stripping cycles, ultimately causing low Coulombic efficiency (CE) and poor cycling performance. These issues are exacerbated under low-temperature conditions due to sluggish kinetics[1–4]. Therefore, their practical application, especially under low-temperature conditions, faces several challenges, particularly concerning the choice of electrolytes[5–8]. The electrolyte plays a crucial role in the performance and safety of LMBs, influencing factors such as ionic conductivity, SEI formation, and overall electrochemical stability[9–12].

Recent advancements have identified ether-based weakly solvating solvents, such as diethyl ether (DEE), as preferred choices for low-temperature LMB electrolytes due to their excellent reduction

[1]Department of Chemical Engineering, Tsinghua University, Beijing, China. [2]Tsinghua University Hefei Institute for Public Safety Research, Hefei, China. [3]State Key Laboratory of Space Power-Sources, Shanghai Institute of Space Power-Sources, Shanghai, China. [4]Xinyuan Qingcai Technology Co., Ltd, Beijing, China. [5]These authors contributed equally: Weili Zhang, Yang Lu, Qingqing Feng. ✉e-mail: zhangweili@tsinghua-hf.edu.cn; liukai2019@tsinghua.edu.cn

 

stability and compatibility with lithium metal. More importantly, DEE offers advantages like rapid de-solvation and the formation of anion-derived SEI. Previous studies have shown that inorganic-rich SEI has unique advantages in inhibiting the growth of lithium dendrites due to its effective blocking of electron tunneling, promoting uniform Li+ transport and good mechanical properties. However, its low dissociation degree of lithium salts results in low ionic conductivity, and its poor high-voltage stability limits compatibility with high-voltage positive electrodes[6,13–15]. Conversely, strongly solvating solvents like dimethoxyethane (DME) exhibit high lithium salt dissociation, allowing DME-based electrolytes to achieve high ionic conductivity, especially at low temperatures. In addition, they can address high-voltage issues with the help of specific additives. Nevertheless, strong solvents face challenges such as difficult de-solvation and large voltage drops, which can lead to uncontrolled growth of lithium dendrites at low temperatures, and will ultimately pose safety risks[16–18]. Many recent reports on low-temperature electrolytes focus on combining strongly and weakly solvating solvents to balance ion conductivity and de-solvation ability. However, due to the strong coupling between ion transport and de-solvation, it is challenging to simultaneously attain high ion conductivity and low de-solvation energy[19,20]. Moreover, The utilization of electrolyte additives represents the most pragmatic and cost-effective strategy for enhancing the low-temperature performance of LMBs. Conventional lithium battery additives such as Fluoroethylene Carbonate (FEC), Lithium Difluorophosphate (LiDFP), Lithium nitrate (LiNO$_3$), 1,3-Propanesultone (PS), etc.[5,7,21–26] have been reported to enhance the stability of the electrode/electrolyte interphase and reduce interfacial impedance by modifying its structure and composition, thereby improving the low-temperature performance of LMBs. Despite significant progress in these low-temperature additives, current formulations remain functionally limited, failing to simultaneously address the multifaceted challenges of LMBs under ultra-low-temperature conditions, including SEI instability, sluggish ion transport kinetics, inhomogeneous lithium deposition, and compromised electrolyte fluidity. Here, we chose a binary solvent system consisting of a mixture of strong (DME) and weak solvating solvents (DEE) for the electrolyte. We demonstrated that introducing a small amount of a strongly solvating solvent into a weakly solvating solvent significantly enhances the ionic conductivity of the electrolyte without notably altering the solvation structure of the weak solvent (DEE: DME = 9:1 vol%). Furthermore, we designed and synthesized a multifunctional additive, perfluoroalkylsulfonyl quaternary ammonium nitrate (PQA-NO$_3$, note as PN), containing both cations (PAQ+) and anions (NO$_3^-$). Based on the frontier orbital theory and the calculation results of reaction energy, PAQ+ can be preferentially reduced on the surface of Li metal to form LiF with high interface energy and Li$_3$N, Li$_2$O, Li$_2$S with high ionic conductivity, ensuring rapid transport of Li+ through SEI and inhibiting the growth of lithium dendrites at ultra-low temperatures. In addition, NO$_3^-$ enters the Li+ solvation shell and repels the solvent, weakening the interaction between Li+ and the solvent, accelerating the process of Li+ de-solvation, and greatly alleviating the inevitable increase in de-solvation energy barrier caused by the introduction of strong solvents DME. Besides, NO$_3^-$ constructs a high oxidation stability positive electrode/electrolyte interface with poor-solvent and rich anions, enabling the ether-based solvent system to match the NMC811 positive electrode for stable cycling at a high operating voltage of 4.3 V. As a result, the Li||NMC811 full cell using the designed electrolyte exhibits stable long-term cycling performance under extremely low-temperature conditions of −60 °C. The actual industrialized pouch cell can stably discharge at −85 °C, maintaining 48.1% of its room temperature capacity and achieving a specific energy of 171.8 Wh kg$^{-1}$ (except tabs and packing foil, same hereafter) at −85 °C. The pouch cell can discharge at a high rate of 3.0 C at −50 °C, achieving a high-specific power of 938.5 W kg$^{-1}$.

## Results

According to molecular frontier orbital theory, electrolyte components with lower LUMO energy levels are thermodynamically more inclined to undergo reductive decomposition. Figure 1a calculates the molecular orbital energy levels of different components in the electrolyte, and the results show that the PN additive molecules have lower LUMO energy levels, leading to preferential decomposition on the lithium metal surface and participation in SEI formation. Furthermore, we immersed lithium metal in a DME solution containing 0.1 M PN for 2 h (detailed experimental results for different immersion times are presented in Supplementary Note 1), and then tested its surface composition using X-ray photoelectron spectroscopy (XPS). Commercial lithium metal foil typically exhibits a native passivation layer rich in Li$_2$CO$_3$ and LiOH (Fig. 1b–e), which increases the impedance and overpotential of the electrode and also affects the subsequent construction of SEI on the electrode surface[21,22]. However, after soaking, the surface of the lithium metal showed significant amounts of inorganic compounds such as LiF, Li$_2$CO$_3$, Li$_2$O, Li$_2$S, and Li$_3$N, demonstrating that PN can react in situ with lithium metal to form an inorganic-rich SEI film, altering the structure of the native passivation layer. The abundant inorganic components are typically considered to enhance the mechanical strength of the SEI, allowing it to accommodate repeated volume changes and inhibit dendrite growth, thereby promoting stable cycling of the lithium metal at low temperatures[6,8]. We further employed the ab initio molecular dynamic (AIMD) calculations to elucidate the interfacial reaction mechanism between the PN additive and the Li metal in pure DME (Figs. 1f and S5–S7) (Source Data 1) or mixed ether-based solvent systems (DEE: DME = 9:1 vol%) (Fig. S8) (Source Data 2)[8,27,28]. Figures 1f and S5–S7 show snapshots of AIMD simulations at different simulation timescales. PN was found to automatically adsorb to the surface of lithium metal, and the NO$_3^-$ anions decompose first, forming Li$_3$N and Li$_2$O components. At the same time, the S=O bond of PAQ+ breaks and generates Li$_2$S components on the surface of lithium metal. As the reactants were exposed to more Li$^0$ by diffusion, the PN underwent a rapid defluorination process via C−F cleavage, leading to a substantial amount of LiF formation. However, the ether solvent is relatively stable with the Li metal, and no decomposition reaction occurs on the lithium metal surface in the simulated time scale. We also observed the same results in the mixed-solvent system (Fig. S8). Therefore, the simulation results demonstrate that PN can preferentially undergo in situ chemical reactions with lithium metal over the solvent, resulting in the formation of an inorganic-rich SEI layer with strong mechanical strength and rapid lithium-ion conduction capability, consistent with the experimental observations.

Additionally, the ion transport properties, de-solvation ability, and film-forming characteristics of 1.0 M LiFSI/DEE (note as DEE), 1.0 M LiFSI/DEE + DME DEE: DME = 9:1 vol%, note as DDE, and DDE + 0.1 M PN (DDE-PN) electrolytes were investigated and compared. As shown in Fig. 2a, the ionic conductivity of the electrolyte measured using a single DEE solvent is relatively low, with a value of only 0.22 mS cm$^{-1}$ at −40 °C and 0.02 mS cm$^{-1}$ at −60 °C, due to insufficient dissociation of lithium salts by the weak solvent. However, the introduction of 10% by volume of the strong solvent DME increases the conductivity of the entire electrolyte system by an order of magnitude at −40 °C and −60 °C, with the DDE electrolyte achieving a conductivity of 2.11 mS cm$^{-1}$ at −40 °C and 0.51 mS cm$^{-1}$ at −60 °C. The presence of 0.1 M PN additive has no significant impact on the conductivity of the mixed-solvent electrolyte system, with the DDE-PN electrolyte exhibiting a high conductivity of 2.32 mS cm$^{-1}$ at −40 °C and 0.62 mS cm$^{-1}$ at −60 °C. Further classical molecular dynamics (MD) simulations were conducted (Fig. S9), utilizing radial distribution functions to describe the average local solute-solute interaction environment (Fig. 2d–f). The analysis indicates that the DEE electrolytes exhibit a characteristic contact ion pair structure, where the Li+ solvation shell contains

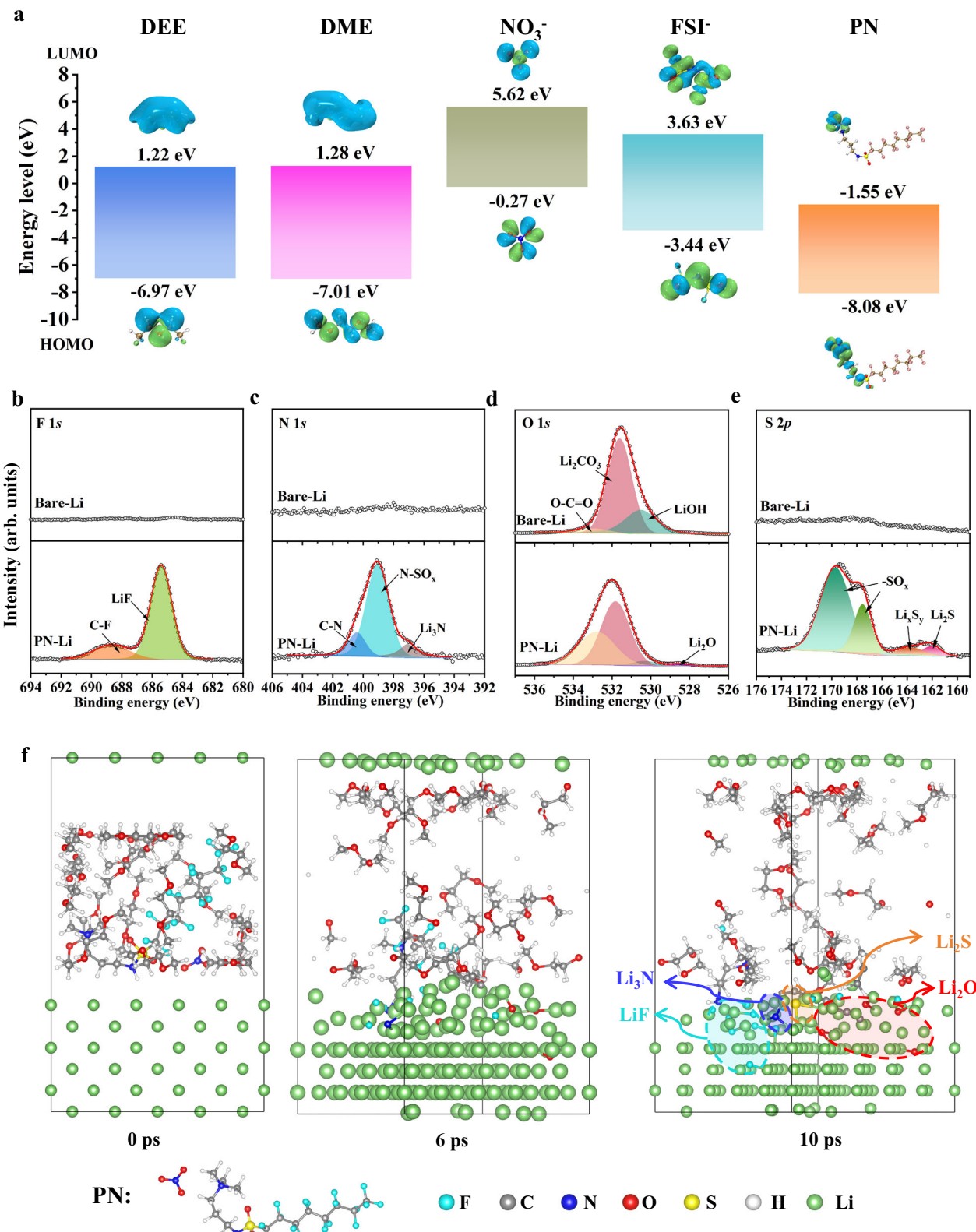

**Fig. 1 | Simulation and characterization of in situ reaction between PN additive and lithium metal. a** Calculated HOMO and LUMO of the solvents, anion, and additives. HOMO and LUMO visualizations are molecular orbital isosurfaces derived from quantum chemical calculations, illustrating the spatial distribution of the wavefunction amplitude, with green and blue indicating its positive and negative phases, respectively. **b** F 1*s*, **c** N 1*s*, **d** O 1*s*, **e** S 2*p* XPS spectra of bare-lithium metal and immersed lithium metal in a DME solution containing 0.1 M PN.

**f** Snapshots from AIMD simulation of decomposition reaction processes between PN with Li metal. The cyan spheres represent fluorine (F) atoms, the gray spheres represent carbon (C) atoms, the blue spheres represent nitrogen (N) atoms, the red spheres represent oxygen (O) atoms, the yellow spheres represent sulfur (S) atoms, the white spheres represent hydrogen (H) atoms, and the green spheres represent lithium (Li) atoms. The initial structure of the AIMD simulation is provided as Supplementary Data 1. Source data are provided as a Source Data file.

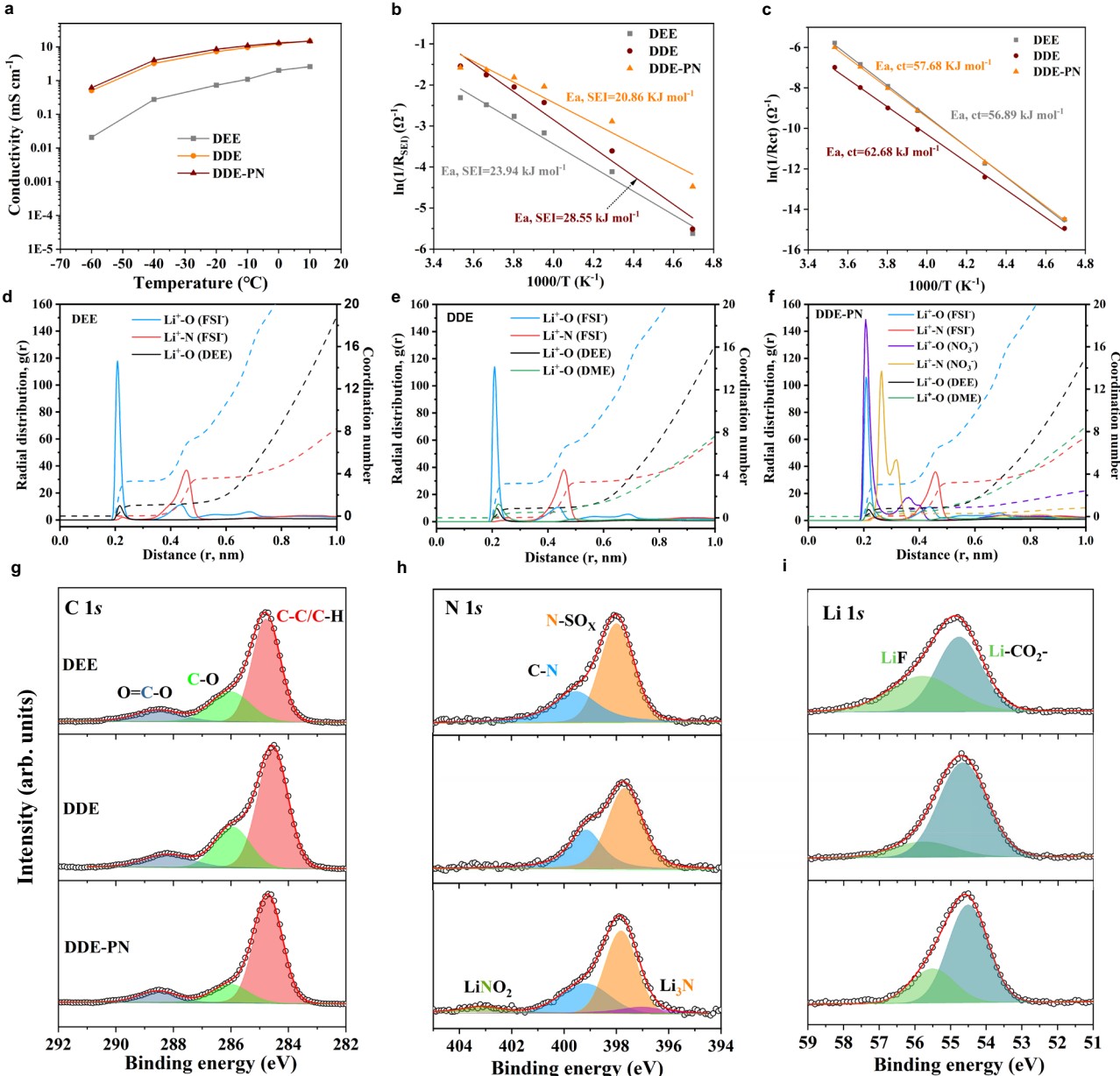

**Fig. 2 | Ion diffusion, charge transfer, solvation structures of electrolytes, and chemical compositions of SEI. a** Ion conductivity of different electrolytes in a wide temperature range. **b** The activation energy of Li⁺ transport in SEI. **c** The activation energies for Li⁺ de-solvation at the negative electrode interface. Radial distribution functions and coordination numbers in **d** DEE, **e** DDE, and **f** DDE-PN. **g** C 1*s*, **h** N 1*s*, **i** Li 1*s* XPS spectra of SEI formed on the surface of Cu were measured after depositing lithium metal with a fixed capacity of 2 mAh cm⁻² at a current density of 1 mA cm⁻² on Cu in Li||Cu coin cells containing DEE, DDE, and DDE-PN electrolytes after 20 cycles. Source data are provided as a Source Data file.

FSI⁻-rich anions with little DEE molecules, with an average coordination number of 3.2 FSI⁻ and 1.0 DEE, consistent with previous studies[6]. With the introduction of the DME solvent, due to the strong interaction between Li⁺ and DME, DME enters the Li⁺ solvation shell, replacing some DEE solvent molecules. The average coordination number becomes 3.2 FSI⁻, 0.88 DEE, and 0.48 DME, indicating that DME entered the solvation shell and squeezed out some DEE solvents. However, when the PN additive is introduced into the mixed-solvent system, NO₃⁻ enters the Li⁺ solvation shell due to its stronger binding ability with Li⁺, displacing the solvent. The average coordination number becomes 3.0 FSI⁻, 0.46 NO₃⁻, 0.58 DEE, and 0.4 DME. The overall decrease in solvents that appeared in the Li⁺ solvation shell effectively alleviates the adverse impact of DME on de-solvation kinetics. By evaluating the binding energies of Li⁺(solvent)ₙ complexes,

the binding energies of the Li⁺(DEE)₁, Li⁺(DEE)₀.₈₈(DME)₀.₄₈, and Li⁺(DEE)₀.₅₈(DME)₀.₄ complexes were determined to be 219.19 kJ mol⁻¹, 316.81 kJ mol⁻¹, and 238.01 kJ mol⁻¹, respectively. Therefore, although the introduction of DME significantly increases the de-solvation barrier for lithium ions, the presence of NO₃⁻ in DDE-PN substantially mitigates the adverse effects caused by strong solvation solvents. The kinetics of different interfacial processes were studied using temperature-variable electrochemical impedance spectroscopy (EIS)[29,30]. The energy barrier of the de-solvation process depends on the number and coordination ability of solvent molecules in the Li⁺ solvation shell. By measuring the charge transfer resistance of Li||Li symmetric cells at different temperatures, the de-solvation activation energy $E_{a,ct}$ of Li⁺ in different electrolytes was determined. Figure 2c shows that the activation energies for the DEE system and the DDE

binary system are 56.89 kJ mol$^{-1}$ and 62.68 kJ mol$^{-1}$, respectively. The higher de-solvation energy in the DDE system is due to the stronger complexation ability of DME with Li$^+$, significantly increasing the difficulty of Li$^+$ de-solvation. However, the PN additive introduces NO$_3^-$ ions, which can enter the first solvation shell, altering the Li$^+$ solvation structure and weakening the Li$^+$-solvent interactions (E$_{a,ct}$ = 57.68 kJ mol$^{-1}$ for the DDE-PN), consistent with the MD results.

To study the SEI formation mechanism, XPS was employed to characterize the composition and structure of the SEI on the cycled lithium metal surface[31–33]. The peak intensity of C–O and C=O in the SEI formed in DEE is significantly lower than that in DDE (Fig. 2g), indicating that solvent decomposition in DEE is suppressed compared to that in DDE. Therefore, in DEE, the SEI mainly consists of a large amount of inorganic substances generated by anion reduction, while the introduction of DME disrupts the anion-enriched solvation structure in the weak solvent, greatly increasing the organic content in the SEI (Fig. 2h, i). Interestingly, the SEI formed in the DDE-PN electrolyte contains more inorganic components such as LiF and Li$_3$N, because the inorganic-rich SEI derived from the decomposition of PN on the lithium metal surface effectively inhibits the solvent decomposition in the mixed binary solvent system. The SEI structure was further studied by XPS spectra at different etching depths. As shown in Figs. S10–S14, in all three electrolytes, the SEI exhibits fewer organic components and more inorganic components with increasing sputtering depth. However, the inner layer of the SEI formed in DDE-PN has significantly more inorganic components (such as LiF, Li$_3$N, Li$_2$S, etc.). This gradient double-layer SEI structure, where the organic layer is located at the top and the inorganic layer is rich close to the lithium surface, has been proven to be an ideal SEI structure for lithium metal[10,34]. Similarly, the kinetic process of interface Li$^+$ transport through SEI was evaluated using variable temperature EIS. As shown in Fig. 2b, in the DEE system, the activation energy for Li$^+$ transport through the SEI (E$_{a,SEI}$) is 23.94 kJ mol$^{-1}$, ~5 kJ mol$^{-1}$ lower than that of the mixed-solvent system, DDE. This is because the weakly coordinating solvent DEE can form an anion-enriched solvation structure, promoting the formation of an anion-derived, inorganic-rich SEI. The inorganic substances dispersed in the anion-derived SEI create abundant phase boundaries and vacancies, facilitating rapid Li$^+$ diffusion and significantly reducing the energy barrier. In contrast, the E$_{a,SEI}$ of DDE-PN decreases to 20.86 kJ mol$^{-1}$, indicating that the SEI derived from the PN additive reduces the Li$^+$ diffusion barrier, even lower than that from DEE. This has greatly accelerated the Li-ion diffusion kinetics, which should be attributed to the in situ reaction of the PN additive with lithium metal, forming LiF, an electronic insulator, and Li$_3$N, a fast ion conductor. This heterogeneous synergistic effect is considered an effective strategy to achieve rapid Li$^+$ transport[34,35]. Therefore, the PN additive allows the DDE binary solvent to combine the advantages of each component while getting rid of their disadvantages: the DDE-PN exhibits better kinetics at the interface (high-quality SEI and rapid de-solvation process) inherited from weak solvents while showing high ionic conductivity in the electrolyte bulk solution brought by the strong solvents.

To test the long-term cycling stability of Li metal in different electrolytes, Li∥Li symmetric cells were assembled. It was found that the DDE-PN electrolyte effectively extended the life of the Li∥Li symmetric cells (Figs. 3a and S15). Especially at high current densities, the DDE-PN electrolyte exhibited the lowest overpotential, indicating better lithium metal reversibility. Lithium metal deposition morphology at different current densities (Figs. 3b–d, S16 and S17) showed that lithium deposition in the DEE system displayed uniformly distributed blocky lithium with minimal gaps at low current densities. However, the introduction of DME resulted in lithium metal depositing in smaller particles with larger gaps, significantly increasing the contact area between lithium metal and the electrolyte and enhancing interfacial side reactions. In contrast, lithium deposition in the DDE-PN

electrolyte displayed a smooth, film-like surface morphology, considered as the preferred one for the long-term cycling of lithium metal. More importantly, as the current density increased, lithium metal in the DEE and DDE electrolyte systems tended to deposit as loose small particles with significantly larger gaps (Fig. 3b, c). Lithium metal in the DDE-PN electrolyte still formed dense (Fig. 3d), compact aggregates even at a high current density of 10 mA cm$^{-2}$, closely related to the good lithium-ion transport kinetics of the DDE-PN system. Under identical deposition capacities, the corresponding thicknesses of lithium deposits in DEE, DDE, and DDE-PN electrolytes measured 18 μm, 21 μm, and 17 μm, respectively. The notably looser and more porous structure observed in DDE electrolytes highlights that the incorporation of DME amplifies parasitic reactions between the electrolyte and lithium metal under fast-charging conditions. In contrast, the DDE-PN electrolyte exhibited the thinnest and most compact lithium deposition morphology, demonstrating its effective capability to suppress interfacial side reactions and promote dense lithium growth under high-rate electrochemical conditions.

We further assembled Li∥Cu cells to study the effects of different temperatures on lithium deposition behavior. At room temperature, the Li∥Cu cells exhibit a notable CE of 99% during the first 80 cycles in three different electrolytes, as shown in Fig. S18. However, with extended cycling, the DDE-PN electrolyte demonstrates the most stable cycling performance, maintaining a high CE of 99.1% (the average value of 100 to 200 cycles). This result proves the high compatibility of the DDE-PN with lithium metal and the long-term reversibility of lithium deposition/stripping. Subsequently, we investigated the electrochemical performance of lithium metal at low temperatures. As shown in Figs. 3e and S19, the CE values of the DEE, measured by Aurbach's method, were 98.5% at 25 °C and 97.2% at −60 °C, respectively. For the DDE binary system, although the ionic conductivity of the electrolyte was high at −60 °C as discussed before, the CE was measured to be only 71.2% at −60 °C, which should be ascribed to the increased de-solvation energy barrier and unstable SEI caused by the introduction of strong solvent, DME. However, the DDE-PN electrolyte exhibited the lowest overpotential, smooth lithium deposition/stripping curves, and stable operating voltage at −60 °C, achieving a CE of 97.5%. This demonstrated that the DDE-PN electrolyte had excellent Li deposition/stripping reversibility at extremely low temperatures, attributed to good interfacial transport kinetics properties. The Li∥Li symmetric cell at −60 °C, as shown in Fig. 3f, short-circuited after ~150 h of Li plating/stripping with the DEE electrolyte. The Li∥Li cell with the DDE is short-circuited at the initial stage of plating/stripping. In contrast, the Li∥Li cell with the DDE-PN stably cycled for over 300 h. This improvement is due to enhanced ionic conductivity, reduced de-solvation energy barriers, and the construction of high-quality SEI, which significantly improved the long-term cycling performance of lithium metal at extremely low temperatures. Additionally, the surface morphology of lithium metal was characterized using scanning electron microscopy (SEM). At −60 °C and a fixed capacity of 1 mAh cm$^{-2}$, lithium metal in the DEE electrolyte maintained uniform deposition without dendrite formation (Fig. 3g). However, in the DDE electrolyte, the Li metal surface showed a large number of unevenly distributed mossy and needle-like dendrites, consistent with previous studies. Cross-sectional SEM images further reveal that compared to the DEE electrolyte, lithium deposition in DDE predominantly exhibits dendritic growth extending from the substrate bottom toward the top surface (Fig. 3h). The difficulty in de-solvation led to a substantial increase in local charge transfer impedance, naturally driving Li deposition kinetics in a tip-driven manner[6,8]. The introduction of the PN additive into the DDE electrolyte resulted in oval-shaped, dendrite-free lithium metal surfaces, and dense deposition structure at −60 °C (Fig. 3i). This demonstrated the good compatibility between the DDE-PN electrolyte and the lithium metal, especially at extremely low temperatures.

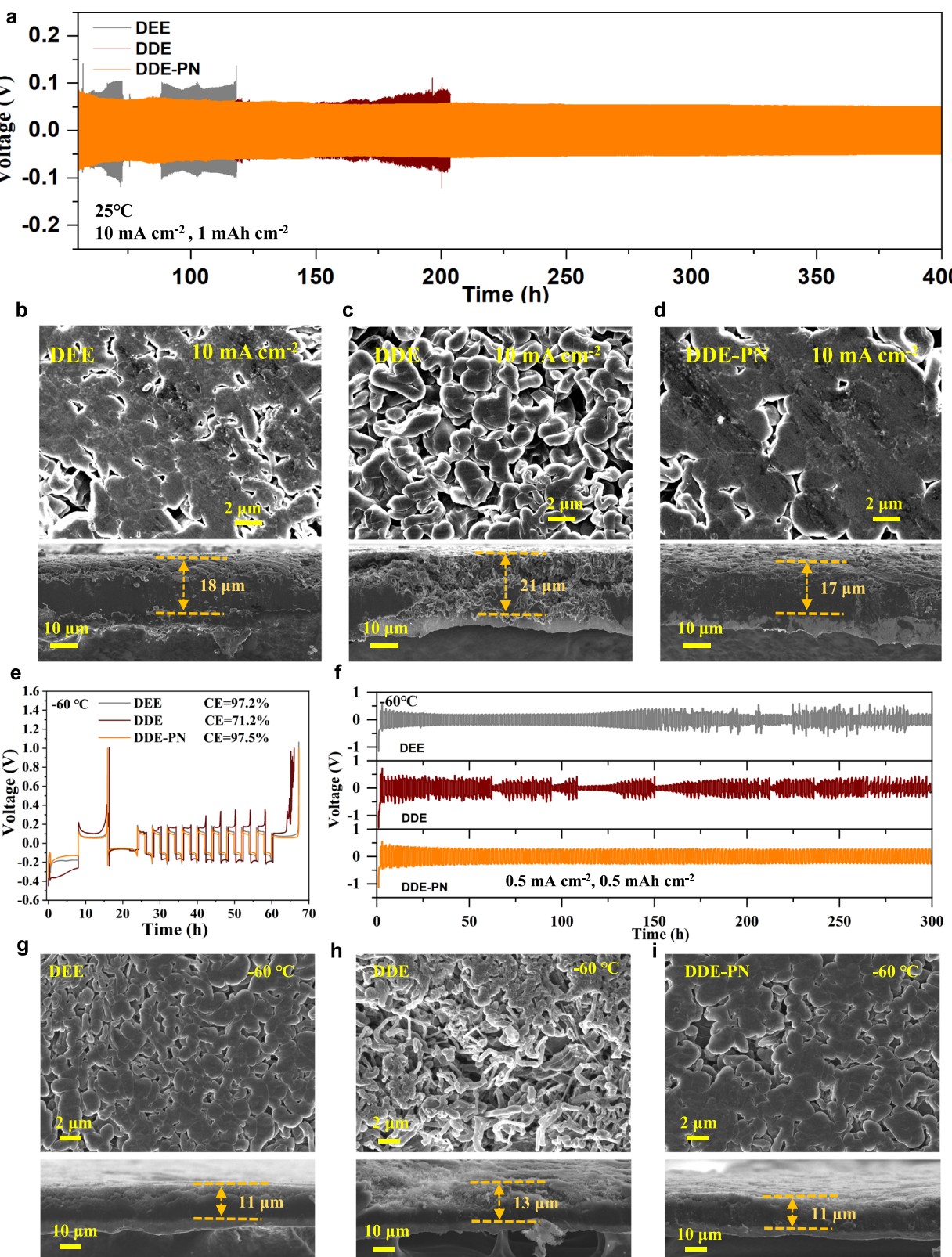

**Fig. 3 | The electrochemical performance of lithium metal under fast-charging and low-temperature conditions. a** Li plating/stripping stability of Li‖Li symmetric cells with a capacity of 1 mAh cm⁻² at 10 mA cm⁻² under 25 °C. The SEM morphologies and cross-section SEM images of Li deposited on Cu foil cycled in **b** DEE, **c** DDE, and **d** DDE-PN electrolytes under 25 °C. After 5 Li deposition/stripping cycles, a fixed capacity of 2.0 mAh cm⁻² was deposited on the Cu at a current density of 10.0 mA cm⁻². **e** Li plating/stripping CE evaluated by Li‖Cu half cells under −60 °C. The testing procedure is as follows: First, deposit at 0.5 mA cm⁻² for

8 h, then fully strip at 0.5 mA cm⁻². Next, deposit again at 0.5 mA cm⁻² for 8 h, followed by stripping/depositing with 1 mAh cm⁻² at 0.5 mA cm⁻² for 9 cycles. Finally, fully strip. **f** Li plating/stripping stability of Li‖Li symmetric cells cycled in different electrolytes with a capacity of 0.5 mAh cm⁻² at 0.5 mA cm⁻² under −60 °C. SEM morphologies and cross-section SEM images of Li deposited on Cu foil cycled in **g** DEE, **h** DDE, and **i** DDE-PN electrolytes under −60 °C with a fixed discharge capacity of 1.0 mAh cm⁻² at 0.5 mA cm⁻² after 5 cycles. Source data are provided as a Source Data file.

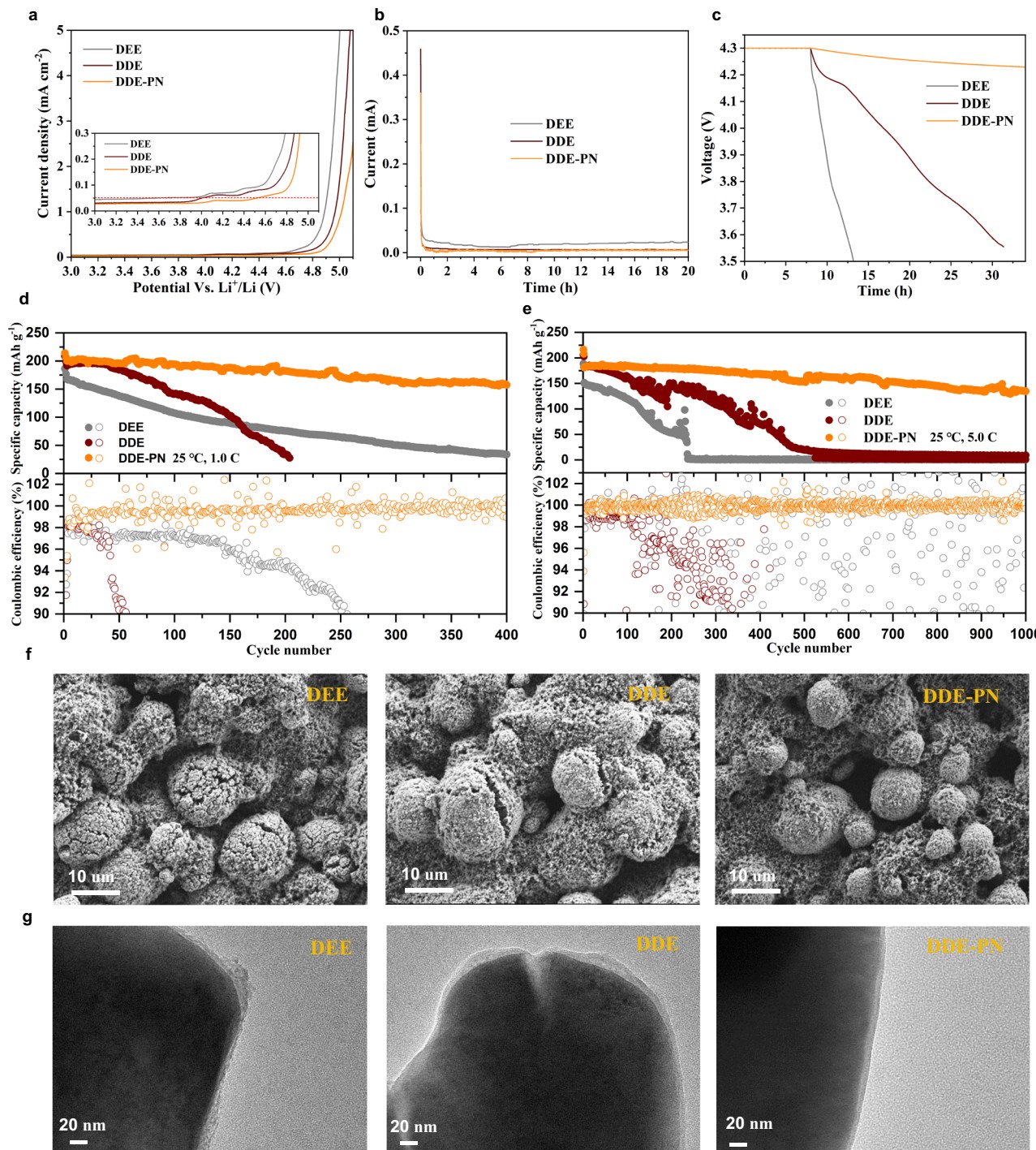

**Fig. 4 | Evaluation of electrolyte oxidation stability and characterization of NCM811 structure. a** Oxidative stability measured via LSV for Li‖Al cells. Scan rate: 5 mV s⁻¹, 25 °C. **b** Typical current relaxation curves collected from Li‖NMC811 half cells during a potentiostatic hold at 4.3 V vs. Li⁺/Li under 25 °C. **c** Self-discharge tests after a potentiostatic hold at 4.3 V vs. Li⁺/Li under 25 °C. Long-term cycling performance in Li‖NMC811 cells at **d** 1.0 C and **e** 5.0 C under 25 °C. 1.0 C = 200 mA g⁻¹. **f** SEM and **g** TEM images of NCM811 electrode in a fully discharged state cycled with different electrolytes after 50 cycles at 1.0 C under 25 °C. Source data are provided as a Source Data file.

A major challenge hindering the application of ether-based dilute solution electrolytes is the poor oxidation stability at high voltages (<4.0 V vs. Li⁺/Li ), which deteriorates the compatibility of ether-based electrolytes with high-voltage NMC811 positive electrodes[36–38]. As demonstrated in Fig. 4a, linear sweep voltammetry (LSV) measurements using aluminum foil working electrodes revealed distinct electrochemical stability boundaries for the tested electrolytes. The conventional ether-based electrolytes DEE and DDE exhibited limited oxidation resistance, with their electrochemical windows constrained below 4.0 V (vs. Li⁺/Li at 0.05 mA/cm² cutoff current density). In striking contrast, the formulated DDE-PN electrolyte achieved significant voltage tolerance extension, pushing the anodic stability limit to 4.5 V under identical testing conditions. This stabilization effect was further corroborated by parallel LSV measurements employing carbon-coated aluminum electrodes, where the PN-modified system consistently maintained enhanced voltage endurance (Fig. S20). We

then evaluated the compatibility of the three electrolytes with NCM811-positive electrodes. Firstly, the static leakage current was measured. As shown in Fig. 4b, by holding the positive electrode at 4.3 V over 10,000 s, the leakage currents in DEE and DDE electrolytes gradually decreased and kept stable at about 0.02 mA and 0.008 mA, respectively. In contrast, the leakage current in the DDE-PN electrolyte rapidly decayed and stabilized at only 0.001 mA, indicating a better interfacial electrochemical stability. Secondly, we tested the self-discharge behaviors of the Li‖NMC811 cells in the three different electrolytes. After charging the cells to 4.3 V and resting for 20 h, there were significant differences in the self-discharge behavior of the cells. As shown in Fig. 4c, the cell voltage in the DDE-PN electrolyte system stabilized at 4.21 V after 20 h of rest. However, the cell voltage in the DDE electrolyte dropped to 3.8 V after 20 h, and in the pure DEE electrolyte, the cell voltage quickly dropped below 3.5 V within only 12 h. This indicates that the introduction of the PN additive effectively prevented the decomposition of the ether-based electrolyte at high voltage, thereby suppressing the self-discharge behavior of the cell[39]. Thirdly, Li‖NCM811 cells were cycled with different electrolytes. As shown in Fig. S21, the Li‖NMC811 half-cell with DDE-PN electrolyte demonstrated improved rate performance, with a specific capacity of 142 mAh g$^{-1}$ even at a high rate of 20.0 C. In contrast, the cells with electrolytes without the PN additive showed poor rate performance, with specific capacities dropping below 50 mAh g$^{-1}$ at 20.0 C for both DEE and DDE electrolytes. Similarly, the Li‖NMC811 half-cell with DDE-PN electrolyte exhibited good long-term cycling performance and relatively stable CE. As shown in Fig. 4d, e, the Li‖NMC811 half-cell with DDE-PN electrolyte retained 80% of its capacity after 400 cycles at 1.0 C and 76% after 1000 cycles at 5.0 C, demonstrating reliable fast charge-discharge capability at high voltage. In contrast, the cells with DEE and DDE electrolytes showed rapid decay at both slow and fast charge-discharge rates during the initial cycles, as ether solvent could not withstand such high working voltages. Additionally, the introduction of the PN additive significantly reduced the interfacial impedance and showed a slower increase in impedance after cycling (Figs. S22 and S23), further confirming the effective improvement in electrode/electrolyte interfacial stability. The comparison of CV curves of Li‖NMC811 half-cell after cycling also proves the long-term stability of the positive electrode/electrolyte interface in DDE-PN electrolyte (Figs. S24 and S25). Therefore, the introduction of the PN additive significantly inhibited the decomposition of the ether-based electrolyte at high voltage, thereby markedly enhancing the compatibility of the ether solvent with the high-voltage NMC811 positive electrode. As shown in Figs. 4f and S26, SEM images showed that the NMC811 positive electrode suffered severe fragmentation after 50 cycles in both DEE and DDE electrolytes. Particle breakage can cause the shedding of active materials or poor electronic contact, leading to increased battery polarization, reduced content of active substances, and decreased reversible capacity of the battery[40]. In contrast, the positive electrode particles remained mostly intact after cycling in the DDE-PN electrolyte. The High Resolution Transmission Electron Microscope (HRTEM) images in Figs. 4g and S27 also revealed that the cathode electrolyte interphase (CEI) on the NMC811 positive electrode surface after cycling in the designed electrolyte was thinner and denser, and maintained better secondary particle integrity with no apparent cracks. However, significant cracks were found within the NMC811 particles after cycling in both DEE and DDE electrolytes, indicating that the CEI formed in commercial electrolytes was not dense enough. In addition, positive electrode degradation is typically associated with the dissolution of transition metals (TMs). Through inductively coupled plasma optical emission spectroscopy (ICP-OES) measurements, we observed that compared to the ether-based electrolyte without PN, the dissolution of TMs in the DDE-PN electrolyte was significantly reduced after 50 cycles (Fig. S28), suggesting that positive electrode degradation is effectively suppressed[5,11].

To find out the origin of the improved compatibility of ether-based electrolytes with high-voltage NMC811, In-depth XPS analysis was conducted to examine the chemical composition of the CEI formed on the surface of NMC811 material after 50 cycles in different electrolytes. A new peak at 283.1 eV in the C 1s spectrum, corresponding to the C−X$_{metal}$ bond[41], was observed in DEE and DDE, indicating severe electrolyte decomposition at the NMC811 interface (Figs. 5a and S29). This is attributed to the poor anodic stability of ether-based electrolytes. With the introduction of the PN additive, the C−X$_{metal}$ peak almost disappeared (Fig. 5a), indicating that the side reactions at the positive electrode interface were greatly suppressed. As sputtering progressed, a significant decrease in the C 1s intensity was observed in DDE-PN (Fig. S29), accompanied by a higher content of inorganic compounds such as LiF, Li$_2$S, and Li$_x$S$_y$ within the inner layer of the CEI formed in the DDE-PN electrolyte. Notably, the CEI generated from the DDE-PN electrolyte exhibited the strongest LiF peak, with an internal LiF content exceeding 39.8%, compared to 32% and 2.4% for the CEIs formed from DDE and DEE electrolytes, respectively (Fig. 5d). This gradient distribution from organic to inorganic components within the CEI imparts effective electronic blocking capabilities, reducing electron penetration into the CEI and thus ensuring the cycling stability of the battery. Unfortunately, the inner layer of the CEI formed in DEE revealed a higher content of carbonaceous organic compounds (from solvent decomposition) and fluorinated organic species (due to incomplete decomposition of the FSI$^-$ anion), but lower amounts of inorganic compounds such as LiF and Li$_2$S (Figs. 5a–d and S29–S31). This indicates that the SEI film may be insufficiently stable and possess poor ion transport properties, rendering it unable to effectively prevent highly reactive free solvents from penetrating the CEI and undergoing continuous degradation[42,43].

Considering that the stability of the electrolyte depends on the aggregation state of ions and solvent molecules in the electric double layer (EDL), we further used in situ attenuated total reflectance surface-enhanced infrared absorption spectroscopy (ATR-SEIRAS) to more clearly reveal the evolution of the solvation structure in the interfacial microenvironment during charging/discharging[43,44]. The interface of the working electrode and electrolyte was measured by ATR-SEIRAS in situ during the charging/discharging under the potentiostatic mode. The voltage was settled between 3.0 and 3.6 V to ensure purely capacitive behavior throughout the test and to avoid the influence of irreversible decomposition products of the electrolyte. As shown in Fig. 5a, b, the intensity of the peak at 1588 cm$^{-1}$, corresponding to NO$_3^-$, increased with charging (Fig. S32), indicating that NO$_3^-$ ions preferentially adsorb in the inner Helmholtz layer (IHP). The is probably due to the smaller ionic size of NO$_3^-$ ions, consistent with previous studies[5,45]. Meanwhile, there are several Raman bands in the 800–900 cm$^{-1}$ range. The bands at 821 and 849 cm$^{-1}$ correspond to free ether solvents with different isomers, and the band at 879 cm$^{-1}$ corresponds to Li$^+$-solvated ether solvent. The area ratio of the free solvent peak to the coordination solvent peak reflects the proportion of the bounding state of the solvent to the total volume (Fig. 5e–g). By fitting the relationship between the ratio of coordination ether solvent and the voltage at the positive electrode/electrolyte interface, as shown in Fig. S33, in the electrolyte without the PN additive, most ether molecules at the positive electrode interface exist in a free state, and the proportion of Li$^+$-solvated ether solvents decreases with increasing voltage. Consequently, an EDL composed of a large amount of anodic unstable, free ether solvents will undoubtedly degrade at high voltage. For the DDE-PN electrolyte system, however, the proportion of Li$^+$-solvated ether solvent increases significantly with increasing voltage, indicating that the introduction of NO$_3^-$ allows more ether to participate in the solvation shell, forming an anodic stable EDL capable of withstanding the high-voltage environment at the positive electrode. Therefore, the preferential adsorption of NO$_3^-$ leads to the displacement of the solvent, and Li$^+$ is attracted in large quantities to the

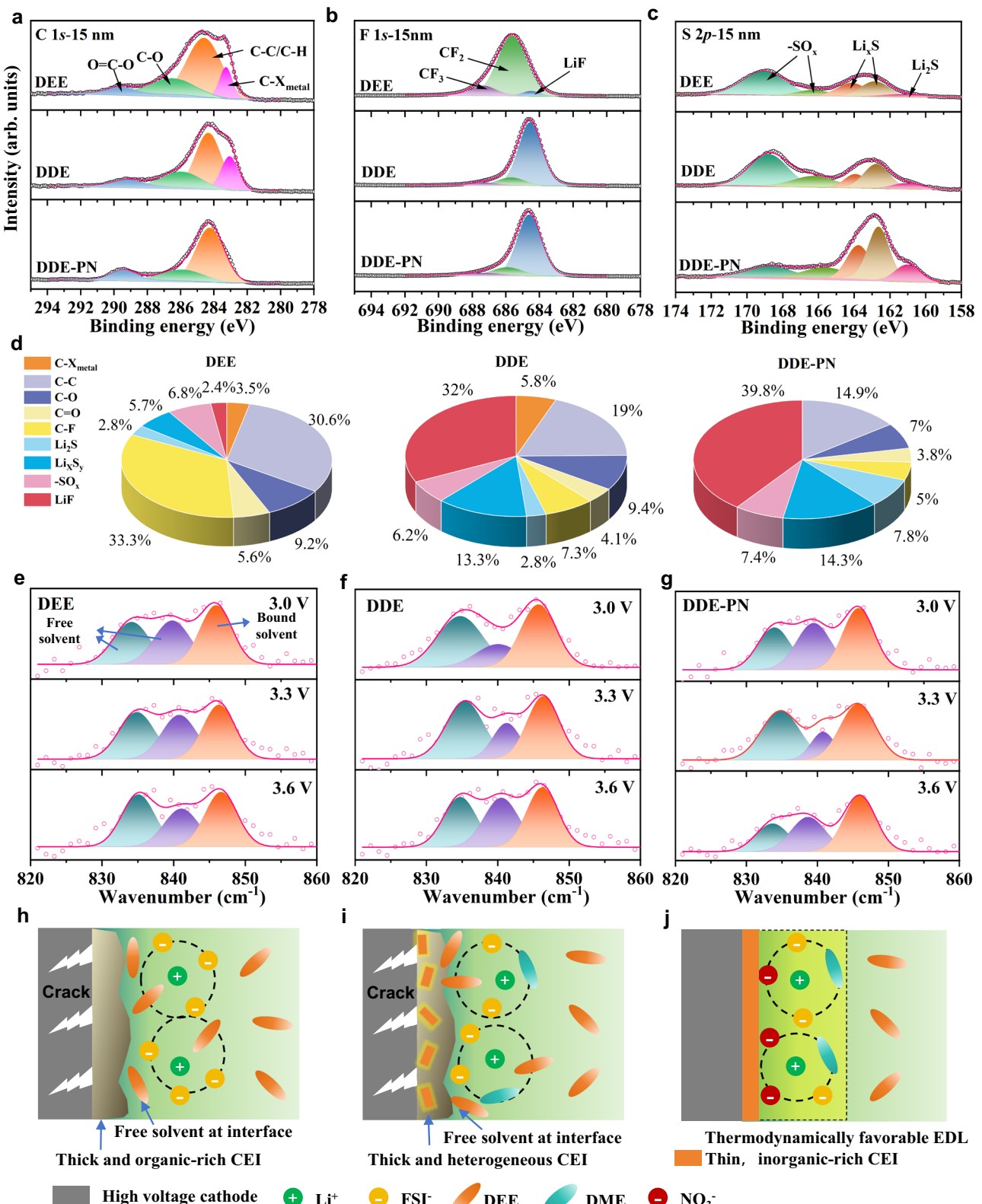

**Fig. 5 | Characterization of CEI compositions and electrolyte structure at the positive electrode surface. a** C 1*s*, **b** F 1*s*, **c** S 2*p* XPS spectra at depth sputtering at 15 nm of NCM811 electrodes in a fully discharged state cycled with different electrolytes after 50 cycles at 1.0 C under 25 °C. **d** Proportion diagram of different components at depth sputtering at 15 nm. ATR-SEIRAS of the surface layer of the working electrode under different voltages in **e** DEE, **f** DDE, and **g** DDE-PN were electrochemically tested using the sweep step function method with a scan rate of 5 mV s⁻¹ and a voltage range of 3.0–3.6 V. Each test voltage was maintained for 2 min at a working temperature of 25 °C. Schematic diagram of the mechanism at the positive electrode/electrolyte interface in **h** DEE, **i** DDE, and **j** DDE-PN. Source data are provided as a Source Data file.

electrode interface due to the strong interaction between $NO_3^-$ and $Li^+$. As a result, the ether present in the EDL layer is extensively bounded with $Li^+$, constructing a highly oxidative-stable EDL. Thus, by reducing the content of active ether solvents in the initially formed EDL, the PN additive is able to alter the stability of ether solvents at the positive electrode interface. Overall, the DDE-PN electrolyte offers a dual protection mechanism by forming a thin, inorganic-rich CEI and establishing a thermodynamically favorable EDL. This effectively mitigates ether-based electrolyte decomposition and positive electrode material structural degradation, significantly enhancing the compatibility of ether-based electrolytes with high-voltage positive electrodes. Therefore, by reducing the content of active ether solvents in the initially formed EDL, the risk of free solvent degradation by the high-voltage positive electrode is minimized. Additionally, this solvent-poor, anion-rich EDL promotes the formation of an inorganic-rich CEI, which, through dual protection, significantly enhances the electrochemical stability of the positive electrode/electrolyte interface in ether-based electrolyte systems (Fig. 5h–j).

Lastly, we investigated the electrochemical properties of these three electrolytes in Li||NMC811 full cells and practical LMBs, respectively. The cycling was conducted under room temperature and ultra-low-temperature conditions. As shown in Fig. 6a, the DDE-PN electrolyte enabled the Li (40 μm)||NMC811 (3.0 mAh cm⁻²) full cell to exhibit good cycling performance at room temperature, with a capacity retention rate of 84% after 300 cycles. In contrast, the DDE and DEE electrolytes showed poor cycling performance, with severe capacity degradation occurring at the initial cycle and after 70 cycles, respectively, due to the suboptimal high-voltage compatibility of ether-based solvents. Additionally, under harsh ultra-low-temperature conditions (−60 °C), the Li||NMC811 full cell was able to stably charge and discharge, providing a high reversible capacity of 110 mAh g⁻¹ over 100 cycles with a capacity retention of 93.3%, indicating stable cycling stability (Fig. 6c). The corresponding charge/discharge curves with temperature variation also exhibited distinct voltage plateaus, indicating good charge/discharge capability at low temperatures (Fig. 6d). Further, 500 mAh Li||NMC811 pouch cells were also evaluated in DDE-PN electrolyte under ultra-low-temperature scenarios. As shown in Fig. 6e, the pouch cells were able to deliver energy densities of 337.3 mAh, 257.4 mAh, and 245.4 mAh at −60 °C, −80 °C, and −85 °C, respectively, corresponding to capacity retention rates of 66.1%, 50.5% and 48.1% relative to room temperature. Excitingly, even at −85 °C, the pouch cell could achieve a specific energy of 171.8 Wh kg⁻¹. The pouch cell with DDE-PN electrolyte could discharge at a high rate of 3.0 C at −50 °C, demonstrating a specific power of 938.5 W kg⁻¹ at such harsh low-temperature conditions (Figs. 6f and S34). Compared to previously reported low-temperature LMBs[6,46–49], our practical pouch cell exhibits significant advantages in specific energy and specific power under extremely low temperatures (Table S2), which is promising to provide reliable power for high-power devices in ultra-low-temperature environments.

## Discussion

In summary, a multifunctional electrolyte additive, PN, was designed and studied for high-power LMBs at ultra-low temperatures. PN preferentially decomposes on the negative electrode to improve the SEI, while optimizing the solvation structure and improving the EDL at the positive electrode/electrolyte interface, which results in well-balanced comprehensive performance, including high-voltage tolerance, high ionic conductivity, low de-solvation energy, and the ability to derive an inorganic-rich SEI. Notably, the practical industrial Li||NMC811 pouch cell cycled in binary ether-based electrolyte, DDE-PN, could stably discharge at −85 °C, maintaining 48.1% of its room temperature capacity, and achieving a specific energy of 171.8 Wh kg⁻¹ at such harsh low-temperature condition. Even more, the pouch cell could discharge at a high rate of 3.0 C at −50 °C, achieving a high-specific power of

938.5 W kg⁻¹. This activation of practical high-rate LMBs under extreme low-temperature conditions provides valuable design insights and theoretical background for low-temperature LMB electrolyte engineering.

## Methods

### Materials

The perfluorooctyl quaternary ammonium iodide was purchased from Shanghai Macklin Biochemical Technology Co., Ltd. The LiFSI, LiNO₃ salt, DEE, and DME solvent were purchased from DodoChem. All the solvents were mixed with molecule sieves to remove the trace water. The electrolytes were prepared in the glove box filled with Ar gas. Metallic Li foil was chased by China Energy Lithium Co., LTD. N-Methyl-2-pyrrolidone (NMP), LiNi₀.₈Mn₀.₁Co₀.₁O₂ (NMC811), LiFePO₄ (LFP), Super-p and PVDF (-50 W) were purchased from Shenzhen Kejing STAR Technology Company. Positive electrode NMC811 laminates were prepared by laying a mixture of 90 wt% NMC811 or LFP particles, 5 wt% Super-p, and 5 wt% PVDF (5.0 wt% NMP) on a carbon-coated aluminum foil current collector (12 μm) with an automatic coating machine (Shenzhen Kejing MSK-AFA-HC100) and dried at 100 °C under vacuum before cell fabrication. The Full cell assembly: NMC811 positive electrode has a capacity of about 3 mAh g⁻¹, a 40 μm thick lithium metal is cut into small discs with a diameter of 14 mm to serve as the negative electrode, with an N/P ratio of -3. Industry-level 500 mAh Li||NMC811 pouch cells (NMC811 Loading: 20.69 mg cm⁻², Thickness: 125 μm, Capacity: 4.0 mAh cm⁻², Lithium metal foil thickness: 50 μm, N/P = 1.26) were purchased from LI-FUN Technology.

Synthesis of PN: 5 g of perfluorooctyl quaternary ammonium iodide was dissolved in 60 mL of acetone, and 1.17 g of silver nitrate was dissolved in 2 mL of deionized water and added dropwise to the above solution. After stirring for 60 min the precipitate was removed by filtration. PN was obtained by cooling crystallization of the filtrate. Purity >99%. The detailed test results of the purity and stability of the synthesized additives and the prepared electrolytes are provided in Supplementary Note 2.

### Electrochemical evaluation

The electrochemical performances of the Li||Cu, Li||Li, Li||NMC811 batteries were examined using 2032-type coin cells (Case and spring material: stainless steel) with a Celgard 2325 separator (Thickness: 25 μm; Lateral dimension: 100 mm; Porosity: 39%; Average pore size: 0.028 μm) conducted on a battery test station (LANHE CT3001A). In order to avoid the corrosion of the stainless steel by the electrolyte, the electrochemical experiments of the coin cells with electrolytes containing LiFSI all use an Al-clad Positive electrode case and an additional piece of aluminum foil underneath the positive electrode disk. The single-side coated positive electrode for the coin cell was cut into circular pieces with a diameter of 12 mm, and each coin cell used 70 μL electrolyte. For the tests of pouch cells, all pouch cells proceeded with a two-cycle formation at 0.1 C and degassed before cycling tests. The stack pressure for pouch cells was -350 kPa. The EIS measurements were carried out under potentiostatic conditions, with a signal amplitude of 10 mV (peak-to-peak) and a frequency range spanning from 0.1 Hz to 100 kHz. The measurements were recorded with 12 data points per decade of frequency to ensure sufficient resolution. Prior to conducting the EIS measurements, the system was allowed to stabilize under an open circuit voltage for a quiet time of 2 s to ensure stable electrochemical conditions. CV (scan rate: 0.1 mV s⁻¹, Voltage range: 3.0–4.3 V) and LSV (scan rate: 5 mV s⁻¹, Voltage range: 3.0–6.0 V) tests were implemented by CHI 760E electrochemical workstation. The Aurbach CE test method was applied to evaluate the CE. Specifically, 5 mAh cm⁻² of Li was deposited on Cu under a current density of 0.5 mA cm⁻² and then charged to 1.0 V to clean the Cu surface. Then plate quantitative Li reservoir ($Q_t$ = 5 mAh cm⁻²) on Cu foil, and repeatedly strip/plate Li with a capacity of 1 mAh cm⁻² ($Q_c$) for n cycles

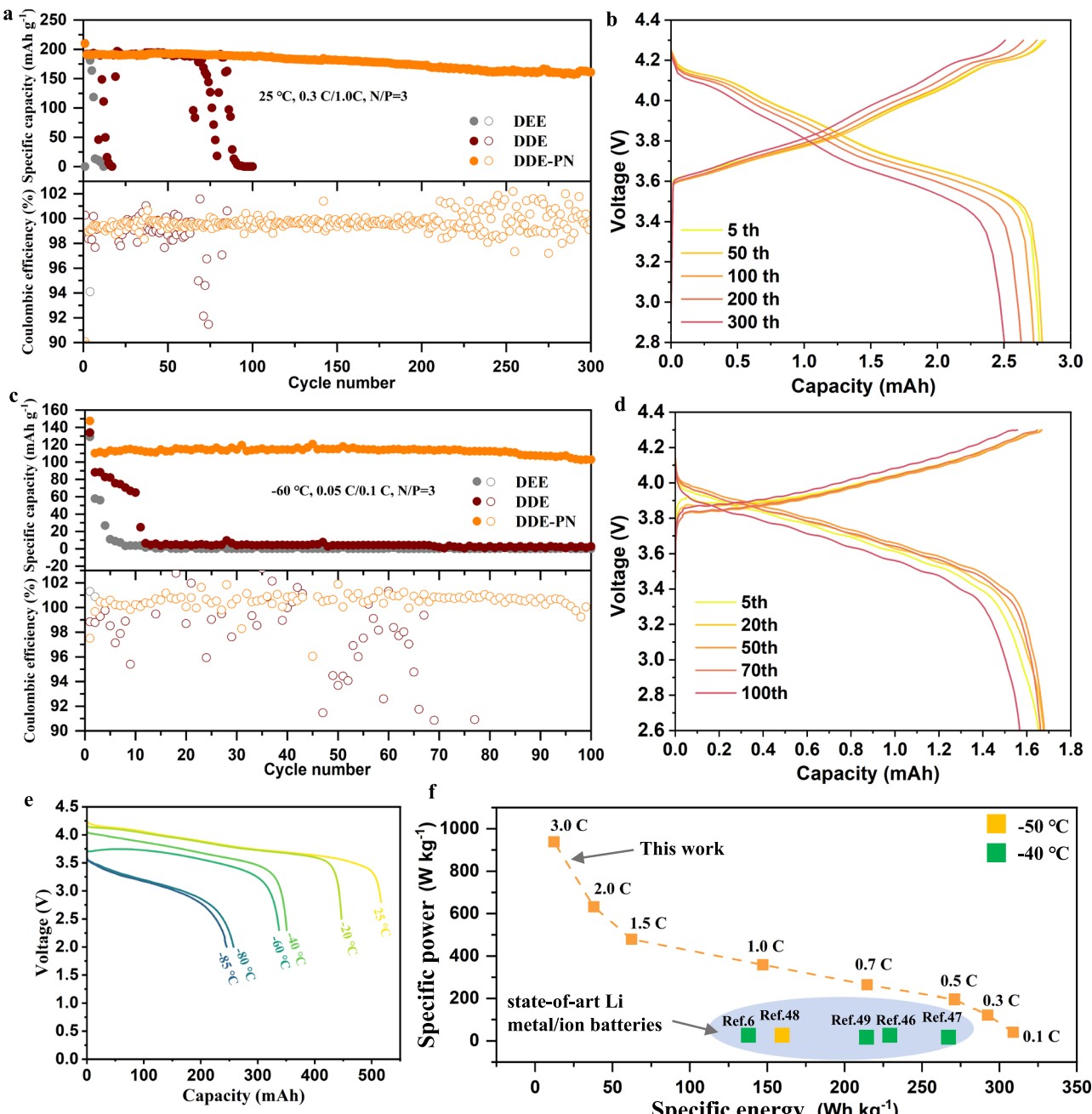

**Fig. 6 | Electrochemical performance of full cell and pouch cell at low temperatures. a** Long-term cycling performances of high-voltage Li‖NMC811 full cells with 40 μm Li metal. The N/P ratio of the Li‖NMC811 cell was 3. The first two formation cycles were carried out at a 0.1 C rate, followed by a 0.3 C charge and 1.0 C discharge. 1.0 C = 200 mA g⁻¹. **b** The corresponding voltage profiles of high-voltage Li‖NMC811 full batteries using electrolytes with DDE-PN. **c** Cycling performance of full cells at −60 °C. **d** The corresponding voltage profiles of high-voltage Li‖NMC811 full batteries using DDE-PN. **e** Discharge profiles (0.1 C, 0.4 mA cm⁻²) of 500 mAh Li‖NMC811 cells using DDE-PN electrolyte at different temperatures. **f** Comparison of a cell-level (output) specific energy and specific power with state-of-the-art electrolytes and our work at ultra-low temperature. Source data are provided as a Source Data file.

($n = 10$) at a current density of 0.5 mA cm⁻², and finally, the remaining Li was stripped to 1.0 V. The average CE could be calculated by following formula (1):

$$CE = \frac{Q_R + nQ_c}{Q_t + nQ_c} \qquad (1)$$

Here $Q_R$ indicates the remaining capacity in the final stripping procedure.

For low-temperature discharge experiments at different temperatures, the pouch cell was charged at a current density of 0.2 C under 25 °C and discharged at a current density of 0.1 C under different temperatures. All cells are placed in a climatic chamber for 2 h to achieve thermal equilibrium before performing electrochemical testing at different temperatures. The specific power $P$ (W kg⁻¹) is obtained by dividing the specific energy $W$ (Wh kg⁻¹) by the discharge time $t$ (s). The ionic conductivity of the electrolyte was measured using a standard 2032 coin cell with two polished 316 stainless steel electrodes placed symmetrically at a set distance. The electrolytic conductivity

value is calculated by the following formula (2):

$$\sigma = \frac{L}{A \times R} \qquad (2)$$

Where $R$ is the resistance, and $A$ and $L$ are the area and spacing between the electrodes, respectively. Data points from 20 °C to −80 °C were measured using VersaStudio software and the symmetrical cells remained at a set temperature controlled by the thermostat for 1 h prior to the test.

## Characterizations

The surface morphology of lithium deposition was characterized by a JSM-7401F SEM. The JEM-2100 Plus transmission electron microscope (TEM) at an accelerating voltage of 200 kV was conducted to characterize CEI layers coating on positive electrode NMC811 particles. The SEI/CEI chemical composition information was measured by using XPS (ESCALAB Xi+) sputtering at different depths. The C1$s$ peak at 284.6 eV was used as the reference for all binding-energy values. The Cu foil or positive electrode in a half-cell after the preparation step was washed with DME to remove residual lithium salts, dried, and then sealed in the glove box until transferred for characterization. The whole process of sample preparation was carried out at room temperature (~25 ± 5 °C) in a glove box with oxygen and water contents below 0.1 ppm. The ionic conductivities of the electrolytes at different temperatures were measured by EIS measurements with two polished 316 platinum plate electrodes symmetrically placed at a set distance in the electrolyte solutions. For the TM dissolution measurement experiment, we disassembled the positive electrode, separator, and negative electrode from the coin cell after 50 cycles and immersed them in 2 mL of DME for 1 h. All liquid organic solutions were collected for acid digestion and further tested using ICP-OES.

## In situ attenuated total reflectance surface-enhanced infrared absorption spectroscopy (ATR-SEIRAS)

ATR-SEIRAS was performed in a sealed three-electrode electrochemical cell. A layer of coarse Au nanoparticles with a thickness of 20 nm was evaporated on the aluminum mesh as the working electrode with surface enhancement effect and lithium as the reference electrode for the working electrode. The electrochemical workstation (CHI 760e) was used to charge and discharge the three-electrode cell, and Raman scanning was carried out after the system reached equilibrium for 2 min at each working potential (OCP vs. Li$^+$/Li, OCP + 0.3 V, OCP + 0.6 V). The Raman system (Horiba HR-800) used in this experiment was equipped with a 785 nm diode laser and a nominal power of 150 mW.

## Computational methods

MD simulations were conducted using the GROMACS software package[50], with atomistic force field parameters for all ions and solvent molecules defined in the AMBER format[51]. Initial energy minimization was performed using the conjugate gradient algorithm with a 0.01 nm step size under periodic boundary conditions in all three dimensions. The Verlet cutoff scheme was applied, with long-range electrostatic interactions handled by the Particle Mesh Ewald method and a cutoff of 1.0 nm for both electrostatic and van der Waals (vdW) interactions. The first equilibration phase was performed under the NPT ensemble with a timestep of 0.0005 ps over 20 ns, where the temperature was gradually increased from 0 K to 298.15 K using a single-step annealing protocol managed by the V-rescale thermostat, and isotropic pressure coupling controlled by the C-rescale algorithm. In the second equilibration phase, the system underwent further equilibration under the NPT ensemble for 20 ns with a 0.001 ps timestep, maintaining a constant temperature of 298.15 K using the V-rescale thermostat, while pressure was regulated using the Parrinello-Rahman barostat. The production run was carried out for 5 ns under the NPT ensemble with a 0.001 ps timestep, keeping the temperature at 298.15 K and pressure at 1.0 bar, utilizing the V-rescale thermostat and Parrinello-Rahman barostat. Representative solvation structures were extracted from the atomistic simulation trajectories and used as initial configurations for quantum chemistry calculations. Density functional theory (DFT) calculations were performed with the Gaussian 16 software[52] at the B3LYP/6-311G** level to optimize the molecular geometries of these solvation structures. Subsequently, binding energies were computed for these optimized structures at the B3LYP/6-311+G(2d,p) level.

The AIMD calculations, based on the DFT as employed in the Vienna Ab initio Simulation Package, are carried out with the Perdew-Burke-Ernzerhof formulation of the generalized gradient approximation[53,54]. DFT-D3 method (Becke-Jonson dampling) is utilized to describe the vdW interaction[55]. We choose the project augmented wave[56] and the plane wave cutoff energy of 400 eV. A $5 \times 5 \times 1$ supercell of cubic phase lithium is cleaved to generate (100) crystallographic plane to represent a positive electrode surface. To avoid interactions between neighboring images, a vacuum region of 20 Å is implemented. 1 PN and 17 DME are inserted in the vacuum space randomly to simulate the electrolyte environment. AIMD simulation for 10 ps at 300 K is performed with a timestep of 1 fs, where the temperature is regulated by NVT ensemble with the Nosé-Hoover thermostat[57,58] and only Γ k-point is set. The 1s1, 2s1, 2s2p2, 2s22p3, 2s22p4, 2s22p5, and 3s23p4 valence electrons are owned by H, Li, C, N, O, F, and S, respectively. The VESTA program[59] is utilized to visualize the crystal structures.

## Data availability
Source data are provided with this paper.

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

## Acknowledgements

We gratefully acknowledge support by the Opening Project of State Key Laboratory of Space-Power Source, the National Key Research and Development Program under Grant No. 2023YFB2503700 (K.L.), the National Science Foundation of China under Grant No. 22071133 (K.L.) and No. 22409117 (W.Z.), Anhui Provincial Natural Science Foundation under Grant No. 240808QB043 (W.Z.), the Tsinghua University-China Petrochemical Corporation Joint Institute for Green Chemical Engineering under Grant No. 224247 (K.L.), Beijing Science and Technology Plan Project under Grant No. Z231100006123003 (K.L.), China Postdoctoral Science Foundation under Grant No. 2024M751747 (K.L.).

## Author contributions

W.Z. and K.L. conceived the idea and designed the experiments. W.Z., Y.L., and Q.F. prepared materials, performed measurements, and analyzed the data. H.W., G.C., H.L., Q.C., Z.L., P.Z., Y.X., W.H., K.Z., and C.D. helped with part of the experiment and data analysis. All authors discussed the results and commented on the manuscript. W.Z. wrote the draft, and K.L. revised and finalized the manuscript.

## Competing interests

The authors declare no competing interests.
