## [Peer Review File · Nature Communications]

Multifunctional electrolyte additive for high power lithium metal batteries at ultra-low temperatures

Corresponding Author: Professor Kai Liu

Version 0:

Reviewer comments:

Reviewer #1

(Remarks to the Author)

This study presents a novel solution to the challenges of sluggish ion transport and dendrite growth in ultra-low temperature lithium metal batteries. By introducing a multifunctional additive, PQA-NO₃, the electrolyte achieves improved interfacial stability, enhanced ion transport, and optimized solvation structure. The proposed system demonstrates excellent low-temperature performance and high power capability, offering a promising approach for high-rate lithium metal batteries in extreme environments. The innovative design of multifunctional additives and its detailed mechanism offer a significant contribution to the field, demonstrating both originality and potential for practical application. The findings are impactful and are expected to arise considerable interest among researchers and professionals working on advanced electrolytes and extreme environment energy storage systems.

So, I recommend publication of this work in Nature Communications after the following minor revisions.

1. For the pouch cells used in this document, more detailed specification parameters need to be provided in the supporting information.
2. It is recommended that the authors supplement their study with AIMD simulations of the in situ reaction process between the PN/mixed solvent solution system and lithium metal, as the final solvent system used in the study is a DME+DEE mixed solvent system.
3. The authors mention transition metal dissolution on page 18, but there is no supporting data in the manuscript. It is suggested that the authors evaluate the dissolution of transition metals from the cathode to demonstrate the positive effect of the PN additive on enhancing the voltage tolerance of the electrolyte.
4. It is recommended that the authors supplement the manuscript with LSV curves using carbon-coated aluminum foil as the working electrode, as FSI is often criticized for corroding stainless steel and pure aluminum current collectors.
5. The manuscript should be further "spell checked" and "grammar checked".

For example:

- ① "week solvent" in line 67 should be "weak solvent"
- ② "defluororination" in line 119 should be "defluorination"

Reviewer #2

(Remarks to the Author)

Reviewer #3

(Remarks to the Author)

Lithium metal batteries (LMBs) face many challenges in ultra-low temperature conditions. An ideal electrolyte for LMBs should not only provide a high Li plating/stripping Coulombic efficiency but also be stable against oxidation at the cathode/electrolyte interface. In this manuscript, the authors designed and synthesized a multifunctional additive, perfluoroalkylsulfonfyl quaternary ammonium nitrate (PQA-NO₃) in ether-based electrolyte, which enables the Li (40 μm)||NMC811 (3 mAh cm⁻²) cells with stable cycling at -60 °C and a 450 Wh kg⁻¹ pouch cell retained 48.1% capacity at -85

°C. The results are interesting, which can promote the development of LMBs at ultra-low temperatures. However, some issues should be addressed before considering publication.

1. In the introduction, please list/compare other electrolyte additives to demonstrate the necessity for newly designed perfluoroalkylsulfonyl quaternary ammonium nitrate (PQA-NO₃).
2. PN can react with Li metal, do they always react until Li or PN is consumed or stop reacting when the SEI is formed? How to control the thickness of the SEI. In the manuscript, the authors immersed lithium metal in a DME solution containing 0.1 M PN for 2 hours, and tested its surface composition using X-ray photoelectron spectroscopy (XPS). What will happen if more hours are applied.
3. The stability of the ether-base electrolyte including PN should also be provided. For example, after synthesizing the electrolyte, does it need low temperature storage? after storing for 3 months or 6 months, is it still stable?
4. In Figure 3, please also provide the cross section SEM images.
5. The authors should double check the data to make sure the data are the same in both manuscript and figures. For example, in Line 134-141 and Figure 2a, the ion conductivity of different electrolytes at -60 degrees are different; in line 243-244 and Figure 3e, the CE value are different.

Reviewer #4

(Remarks to the Author)

In this manuscript, a coupled strategy combining a multifunctional additive with a mixed-solvent electrolyte design was proposed to significantly improve the low-temperature performance of lithium metal batteries. This study offers a profound and insightful explanation of the interfacial chemistry mechanisms attributed to the cationic and anionic components of the additive at the electrode interfaces. Moreover, it presents an innovative and highly effective strategy for low-temperature electrolyte design, culminating in the realization of remarkably competitive low-temperature performance. Sufficient characterization has been done to demonstrate the mechanism of electrolyte design in interfacial chemistry and solvation chemistry. Appropriate figures and descriptions were given to make the paper understood easier. As a result, I recommend it for publication in Nature Communications after the following minor revisions.

1. The presented Figure 2i in the manuscript does not appear to the XPS F-spectrum results as stated. The authors are strongly encouraged to thoroughly review the XPS characterization data for the lithium metal anode, as there seems to be some errors in the data presentation.
2. Additionally, the reviewer recommends including detailed depth-profile analysis data for the S-spectrum and Li-spectrum in the Supporting Information.
3. For Figure S13, the reviewer suggests including specific values for the cycling rate at different cycle numbers to more clearly illustrate the battery's rate capabilities.
4. Starting from line 323 of the manuscript, there are errors in the numbering and presentation of the SEM and TEM images. The authors are urged to carefully review and rectify these inconsistencies to ensure accuracy.
5. In the figure caption for Figure 6, there is no image labeled as Figure 6g. The authors have mistakenly referred to Figure 6f as Figure 6g, and this error must be corrected.

Version 1:

Reviewer comments:

Reviewer #1

(Remarks to the Author)

The authors have well revised the paper. Accept.

Reviewer #2

(Remarks to the Author)

Reviewer #3

(Remarks to the Author)

I am satisfied with the revision made and now recommend the manuscript for publication.

Reviewer #4

(Remarks to the Author)

The author has addressed all my comments. I recommend to publish as it is.

We thank all the reviewers for their valuable comments on our manuscript. Their constructive suggestions for improvement have certainly raised the quality of our manuscript. In order to respond to the reviewers' concerns more clearly, we have addressed their comments point by point.

RESPONSE TO REVIEWERS' COMMENTS

Reviewer #1:

This study presents a **novel** solution to the challenges of sluggish ion transport and dendrite growth in ultra-low temperature lithium metal batteries. By introducing a multifunctional additive, PQA-NO₃, the electrolyte achieves improved interfacial stability, enhanced ion transport, and optimized solvation structure. The proposed system demonstrates **excellent** low-temperature performance and high power capability, offering a **promising** approach for high-rate lithium metal batteries in extreme environments. The **innovative** design of multifunctional additives and its detailed mechanism offer a **significant contribution to the field**, demonstrating **both originality and potential for practical application**. The findings are **impactful and are expected to arise considerable interest** among researchers and professionals working on advanced electrolytes and extreme environment energy storage systems.

So, I recommend publication of this work in Nature Communications after the following minor revisions.

Response: Thank you very much for your positive comments!

Comment 1: For the pouch cells used in this document, more detailed specification parameters need to be provided in the supporting information.

Response to comment 1: Thank you very much for your valuable suggestions.

We have added detailed specification parameters of pouch cells on lines 555-558 on page 30 in the revised manuscript.

Please see lines 544-547 on page 30 in the revised manuscript:

Industry-level 500 mAh Li||NMC811 pouch cells (Cathode [Loading: 20.69 mg cm⁻², Thickness: 125 μm, Capacity: 4.0 mAh cm⁻²], Anode [Lithium metal foil thickness: 50 μm], N/P=1.26) were purchased from LI-FUN Technology.

Comment 2: It is recommended that the authors supplement their study with AIMD simulations of the in situ reaction process between the PN/mixed solvent solution system and lithium metal, as the final solvent system used in the study is a DME+DEE mixed solvent system.

Response to comment 2: Thank you very much for your valuable suggestions.

Following your suggestion, we have supplemented the AIMD simulation experiments of the in situ reaction process between PN and lithium metal in the mixed solvent solution system. The

experimental results show that the simulation results of the mixed ether-based solvent and pure DME solvent systems are consistent. As shown in Fig. S8. PN preferentially undergoes in situ decomposition on the lithium metal surface over the solvent, resulting in inorganic components such as Li_2O , Li_3N , Li_2S , and LiF .

Fig. S8. Simulation of in-situ reaction between PN additive and lithium metal. Snapshots from AIMD simulation of decomposition reaction processes between PN with Li metal in a mixed ether solvent system (DEE: DME=9:1 vol%).

We have added these new data as new Fig. S8 in SI, and the corresponding explanations were added on lines 121-137 on page 7-8 in the revised manuscript.

We further employed the ab initio molecular dynamic (AIMD) calculations to elucidate the interfacial reaction mechanism between the PN additive and the Li metal anode in pure DME (Fig. 1f, Fig. S5-S7) or mixed ether-based solvent systems (Fig. S8).^[8,26,27] Fig. 1f and Fig. S5-S7 show snapshots of AIMD simulations at different simulation timescales. PN was found to automatically adsorb to the surface of lithium metal, and the NO_3^- anions decompose first, forming Li_3N and Li_2O components. At the same time, the $\text{S}=\text{O}$ bond of PAQ^+ breaks and generates Li_2S components on the surface of lithium metal. As the reactants were exposed to more Li^0 by diffusion, the PN underwent a rapid defluorination process via C-F cleavage, leading to a substantial amount of LiF formation. However, the DME solvent is relatively stable with the Li metal, and no decomposition reaction occurs on the lithium metal surface in the simulated time scale. We also observed the same results in the mixed solvent system (Fig. S8). Therefore, the simulation results demonstrate that PN can preferentially undergo in-situ chemical reactions with lithium metal over the solvent, resulting in the formation of an inorganic-rich SEI layer with

strong mechanical strength and rapid lithium-ion conduction capability, consistent with the experimental observations.

Comment 3: The authors mention transition metal dissolution on page 18, but there is no supporting data in the manuscript. It is suggested that the authors evaluate the dissolution of transition metals from the cathode to demonstrate the positive effect of the PN additive on enhancing the voltage tolerance of the electrolyte.

Response to comment 3: Thank you very much for your valuable suggestions.

We have conducted additional experiments on the dissolution of transition metals. Inductively coupled plasma optical emission spectroscopy (ICP-OES) measurements revealed that, compared to the ether-based electrolyte without PN, the dissolution of transition metals in the DDE-PN electrolyte was significantly reduced after 50 cycles (Fig. S28). This demonstrates that the introduction of PN effectively enhances the voltage stability of the electrolyte, significantly suppresses the dissolution and loss of transition metals, and provides robust protection for the high-voltage cathode.

Fig. S28. Characterization of cathode Stability. Transition metal (TM) dissolution measured by inductively coupled plasma mass spectrometry (ICP-MS) after 50 cycles.

We have added these new data as new Fig. S28 in SI, and the corresponding explanations were added on lines 370-375 on page 19 in the revised manuscript.

In addition, cathode degradation is typically associated with the dissolution of transition metals (TMs). Through inductively coupled plasma optical emission spectroscopy (ICP-OES) measurements, we observed that compared to the ether-based electrolyte without PN, the dissolution of TMs in the DDE-PN electrolyte was significantly reduced after 50 cycles (Fig. S28), suggesting that cathode degradation is effectively suppressed.

Comment 4: It is recommended that the authors supplement the manuscript with LSV curves using carbon-coated aluminum foil as the working electrode, as FSI is often criticized for corroding stainless steel and pure aluminum current collectors.

Response to comment 3: Thank you very much for your valuable suggestions.

Following your suggestion, we have incorporated the LSV test using carbon-coated aluminum foil as the working electrode. As demonstrated in Fig. 4a, linear sweep voltammetry (LSV) measurements using aluminum foil working electrodes revealed distinct electrochemical stability boundaries for the tested electrolytes. The conventional ether-based electrolytes DEE and DDE exhibited limited oxidation resistance, with their electrochemical windows constrained below 4.0 V (vs. Li/Li⁺ at 0.05 mA/cm² cutoff current density). In striking contrast, the formulated DDE-PN electrolyte achieved remarkable voltage tolerance extension, pushing the anodic stability limit to 4.5 V under identical testing conditions. This stabilization effect was further corroborated by parallel LSV measurements employing carbon-coated aluminum electrodes, where the PN-modified system consistently maintained superior voltage endurance (Fig. S20).

Fig. S20. Evaluation of electrolyte oxidation stability. Oxidative stability measured via LSV for Li||carbon-coated Al cells.

We have added these new data as new Fig. S20 in SI, and the corresponding explanations were added on lines 313-323 on page 17 in the revised manuscript.

Comment 5: The manuscript should be further “spell checked” and “grammar checked” .

For example:

- ① “week solvent” in line 67 should be “weak solvent”
- ② “defluororination” in line 119 should be “defluorination”

Response to comment 5: Thank you so much for your careful check. We have already reviewed the entire text and made the corresponding revisions.

Reviewer #3 (Remarks to the Author):

Lithium metal batteries (LMBs) face many challenges in ultra-low temperature conditions. An ideal electrolyte for LMBs should not only provide a high Li plating/stripping Coulombic efficiency but also be stable against oxidation at the cathode/electrolyte interface. In this manuscript, the authors designed and synthesized a multifunctional additive, perfluoroalkylsulfonyl quaternary ammonium nitrate (PQA-NO₃) in ether-based electrolyte, which enables the Li (40 μm)||NMC811 (3 mAh cm⁻²) cells with stable cycling at -60 °C and a 450 Wh kg⁻¹ pouch cell retained 48.1% capacity at -85 °C. The results are **interesting**, which can **promote** the development of LMBs at ultra-low temperatures. However, some issues should be addressed before considering publication.

Response: Thank you very much for your positive comments!

Comment 1: In the introduction, please list/compare other electrolyte additives to demonstrate the necessity for newly designed perfluoroalkylsulfonyl quaternary ammonium nitrate (PQA-NO₃).

Response to comment 1: Following your suggestion, we have listed examples of other electrolyte additives in the Introduction to highlight the necessity of the newly designed PQA-NO₃.

Please see lines 67-76 on page 4 in the revised manuscript:

Moreover, The utilization of electrolyte additives represents the most pragmatic and cost-effective strategy for enhancing the low-temperature performance of LMBs. Conventional lithium battery additives such as fluoroethylene carbonate, lithium difluorophosphate, lithium nitrate, 1,3-Propanesultone etc.^[5,7,21-24] have been reported to enhance the stability of the electrode/electrolyte interphase and reduce interfacial impedance by modifying its structure and composition, thereby improving the low-temperature performance of LMBs. Despite significant progress in these low-temperature additives, current formulations remain functionally limited, failing to simultaneously address the multifaceted challenges of LMBs under ultra-low temperature conditions, including SEI instability, sluggish ion transport kinetics, inhomogeneous lithium deposition, and compromised electrolyte fluidity.

Comment 2: PN can react with Li metal, do they always react until Li or PN is consumed or stop reacting when the SEI is formed? How to control the thickness of the SEI. In the manuscript, the authors immersed lithium metal in a DME solution containing 0.1 M PN for 2 hours, and tested its surface composition using X-ray photoelectron spectroscopy (XPS). What will happen if more hours are applied.

Response to comment 2: We sincerely thank the Reviewer for raising this important question. To address this issue, we conducted additional experiments in which lithium metal was immersed in an excess of 0.1 M PN-DME solution for varying durations, followed by XPS analysis to measure the thickness of the SEI layer on the lithium metal surface. As shown in the Fig. S1, when the

immersion time was 1 hour, the lithium metal signal was detected at a sputtering depth of 20 nm. When the immersion time was extended to 2 hours, the lithium metal signal was not appeared until the sputtering depth reaches 50 nm. Further prolonging the immersion time did not significantly increase the SEI thickness, as the lithium metal signal was still detectable at a sputtering depth of 50 nm. Based on these observations, we propose that the formation of a stable SEI layer through the reaction between PN and lithium metal requires a certain duration (≥ 2 hours). Once the SEI layer is pre-formed on the lithium metal surface, it effectively prevents further reactions between the lithium metal and electrolyte components. Therefore, the thickness of the in situ formed SEI layer can be effectively controlled by regulating the resting time after battery assembly.

Fig. S1. XPS characterization of SEI layer thickness. Li 1s XPS depth profiles of immersed lithium metal in a DME solution containing 0.1 M PN for (a) 1h, (b) 2h, and (c) 6h.

We have added these new data as new Fig. S1 in SI, and the corresponding explanation has been added as Note 1 on page 2 of the revised SI.

Comment 3: The stability of the ether-base electrolyte including PN should also be provided. For example, after synthesizing the electrolyte, does it need low temperature storage?
after storing for 3 months or 6 months, is it still stable?

Response to comment 3: We sincerely thank the Reviewer for raising this important question. To assess the long-term storage stability of the synthesized additive, we re-examined the PN additive synthesized 6 months prior through comprehensive ^1H and ^{19}F nuclear magnetic resonance (NMR) spectroscopic analysis. As illustrated in Fig. S2 and Fig. S3, both ^1H and ^{19}F resonance peaks maintained identical chemical shift positions and intensity distributions after prolonged ambient-temperature storage, demonstrating no detectable chemical degradation or structural

evolution. This remarkable spectral consistency conclusively confirms the exceptional shelf stability of the PN additive under routine storage conditions over extended periods.

Regarding the long-term storage stability of ether-based electrolytes containing PN additives, we compared the ^{19}F NMR spectra of a pre-synthesized electrolyte stored for six months with that of a freshly prepared electrolyte (Fig. S4). The results showed that the fluorine NMR peaks of the electrolyte containing PN additives after six months storage under room temperature remained consistent with those of the freshly prepared electrolyte. This demonstrates the stability of the electrolyte during long-term storage at room temperature. Furthermore, to evaluate the aging resistance of the electrolyte system, we subjected both 6-month-aged and freshly prepared electrolyte formulations to rigorous moisture and acid content analysis via Karl Fischer titration and Bromothymol Blue Indicator titration, respectively. As summarized in Table S1, the aged electrolytes exhibited negligible variation in moisture levels and acid content compared to their freshly prepared counterparts after 6 months of storage under ambient conditions. All parameters remain within the industrial acceptance threshold range, collectively confirming the excellent chemical inertness and hydrolytic stability of this electrolyte system during long-term storage.

Fig. S2. Comparison of NMR Spectra of PN Additives. ^1H NMR spectra of PN stored at room temperature for 6 months and fresh PN.

Fig. S3. Comparison of NMR Spectra of PN Additives. ^{19}F NMR spectra of electrolytes containing PN stored at room temperature for 6-month-aged and freshly prepared electrolytes.

Fig. S4. Comparison of NMR Spectra of electrolytes. ¹⁹F NMR spectra of PN stored at room temperature for 6 months and fresh PN.

Table S1. Comparison of acidity and moisture content between long-term stored electrolytes and fresh electrolytes

	Fresh	6 months
Free acid content (ppm)	36	38
moisture content (ppm)	12	10

We have added these new data as new Fig. S2-S4 in SI, and the corresponding explanation has been added as Note 2 on page 3-5 of the revised SI.

Comment 4: In Figure 3, please also provide the cross section SEM images.

Response to comment 4:

Thank you very much for your valuable suggestions. As your suggestion, we have carefully repeated the SEM characterization experiments and added cross section SEM images in Fig. 3.

The corresponding explanations were added on lines 251-259 on page 13 and lines 290-298 on page 15 in the revised manuscript.

lines 251-259 on page 13:

Under identical deposition capacities, the corresponding thicknesses of lithium deposits in DEE, DDE, and DDE-PN electrolytes measured 18 μm , 21 μm , and 17 μm , respectively. The notably looser and more porous structure observed in DDE electrolyte highlights that the incorporation of DME amplifies parasitic reactions between the electrolyte and lithium metal under fast-charging conditions. In contrast, the DDE-PN electrolyte exhibited the thinnest and most compact lithium deposition morphology, demonstrating its exceptional capability to suppress interfacial side reactions and promote dense lithium growth under high-rate electrochemical conditions.

lines 290-298 on page 15:

Cross-sectional SEM images further reveal that, compared to the DEE electrolyte, lithium deposition in DDE predominantly exhibits dendritic growth extending from the substrate bottom toward the top surface. The difficulty in desolvation led to a substantial increase in local charge transfer impedance, naturally driving Li deposition kinetics in a tip-driven manner.^[6,8] The introduction of the PN additive into the DDE electrolyte resulted in oval-shaped, dendrite-free lithium metal surfaces and dense deposition structure at -60 °C. This demonstrated the good compatibility between the DDE-PN electrolyte and the lithium anode, especially at extremely low temperatures.

Fig.3: The electrochemical performance of lithium metal anode under fast charging and low-temperature conditions. (a) Cyclic stability of Li||Li symmetric cells with a capacity of 1 mAh cm^{-2} at 10 mA cm^{-2} under 25°C . The SEM morphologies and cross section SEM images of Li deposited on Cu foil cycled in (b) DEE, (c) DDE and (d) DDE-PN electrolytes under 25°C . (e) Li metal plating/stripping CE evaluated by Li||Cu half cells at 0.5 mA cm^{-2} with a fixed discharge capacity of 1.0 mAh cm^{-1} under -60°C . Cyclic stability of Li||Li symmetric cells with a capacity of 1 mAh cm^{-2} at 0.5 mA cm^{-2} under -60°C . SEM morphologies and cross section SEM images of Li deposited on Cu foil cycled in (g) DEE, (h) DDE and (i) DDE-PN electrolytes under -60°C .

Comment 5: The authors should double check the data to make sure the data are the same in both manuscript and figures. For example, in Line 134-141 and Figure 2a, the ion conductivity of different electrolytes at -60 degrees are different; in line 243-244 and Figure 3e, the CE value are different.

Response to comment 5: Thank you so much for your careful check. We are very sorry for this error. We have carefully checked and corrected this error.

Reviewer #4 (Remarks to the Author):

In this manuscript, a coupled strategy combining a multifunctional additive with a mixed-solvent electrolyte design was proposed to significantly improve the low-temperature performance of lithium metal batteries. This study offers a **profound and insightful explanation** of the interfacial chemistry mechanisms attributed to the cationic and anionic components of the additive at the electrode interfaces. Moreover, it presents **an innovative and highly effective** strategy for low-temperature electrolyte design, culminating in the realization of remarkably competitive low-temperature performance. **Sufficient characterization** has been done to demonstrate the mechanism of electrolyte design in interfacial chemistry and solvation chemistry. **Appropriate** figures and descriptions were given to make the paper understood easier. As a result, I recommend it for publication in Nature Communications after the following minor revisions.

Response: Thank you very much for your positive comments!

Comment 1: The presented Figure 2i in the manuscript does not appear to be the XPS F-spectrum results as stated. The authors are strongly encouraged to thoroughly review the XPS characterization data for the lithium metal anode, as there seems to be some errors in the data presentation.

Response to comment 1: Thank you so much for your careful check. We are very sorry for this error. We have reviewed the entire XPS data and the incorrectly labeled Fig. 2i has been replaced.

Fig.2: Ion diffusion, charge transfer, solvation structures of electrolytes, and chemical compositions of SEI. (a) Ion conductivity of different electrolytes in a wide temperature range. (b) The activation energy of Li⁺ transport in SEI. (c) The activation energies for Li⁺ de-solvation at the anode interface. Radial distribution functions and coordination numbers in (d) DEE, (e) DDE and (f) DDE-PN. (g) C 1s, (h) N 1s, (i) Li 1s XPS spectra of Li metal cycled with different electrolytes.

Comment 2: Additionally, the reviewer recommends including detailed depth-profile analysis data for the S-spectrum and Li-spectrum in the Supporting Information.

Response to comment 2: Thank you very much for your valuable suggestions.

As you requested, we have supplemented the depth-profile analysis data for the S-spectrum and Li-spectrum, as shown in the Fig. S13 and Fig. S14. The inner layer of the SEI formed in DDE-PN has significantly fewer organic components and more inorganic components. This is an important piece of evidence supporting the better electrochemical performance of the lithium metal anode in

the DDE-PN electrolyte.

Fig. S13. XPS characterization of SEI chemical composition. XPS depth profiles of S 2p of SEI formed on Lithium metal surface after 20 cycles in Li||Cu coin cells with DEE, DDE and DDE-PN electrolyte.

Fig. S14. XPS characterization of SEI chemical composition. XPS depth profiles of Li 1s of SEI formed on Lithium metal surface after 20 cycles in Li||Cu coin cells with DEE, DDE and DDE-PN electrolyte.

We have added these new data as new Fig. S13 and Fig. S14 in SI, and the corresponding explanations were added on lines 202-205 on page 10 in the revised manuscript.

As shown in Fig. S10-S14, in all three electrolytes, the SEI exhibits fewer organic components and more inorganic components with increasing sputtering depth. However, the inner layer of the SEI formed in DDE-PN has significantly more inorganic components (such as LiF, Li₃N, Li₂S, etc.).

Comment 3: For Figure S13, the reviewer suggests including specific values for the cycling rate at different cycle numbers to more clearly illustrate the battery's rate capabilities.

Response to comment 3: Thank you very much for your valuable suggestions.

Following your suggestion, we have added the specific values of the cycling rates as a new Fig. S13 in the revised Supplementary Information.

Fig. S13. Electrochemical performance evaluation of cathode. Rate capability of Li||NMC811 cells under different charging/discharging rates.

Comment 4: Starting from line 323 of the manuscript, there are errors in the numbering and presentation of the SEM and TEM images. The authors are urged to carefully review and rectify these inconsistencies to ensure accuracy.

Response to comment 4: Thank you so much for your careful check. We have carefully checked and corrected this error.

Comment 5: In the figure caption for Figure 6, there is no image labeled as Figure 6g. The authors have mistakenly referred to Figure 6f as Figure 6g, and this error must be corrected.

Response to comment 5: Thank you so much for your careful check. We have carefully checked and corrected this error.

Fig.6: Electrochemical performance of full cell and pouch cell at low temperatures. (a) Long-term cycling performances of high-voltage Li||NMC811 full cells with 40µm Li anode. The N/P ratios of the Li||NMC811 cell was 3. The first two formation cycles were carried out at a 0.1 C rate, followed by 0.3 C charge and 1.0 C discharge. (b) The corresponding voltage profiles of high-voltage Li||NMC811 full batteries using electrolytes with DDE-PN. (c) Cycling performance of full cells at -60 °C. (d) The corresponding voltage profiles of high-voltage Li||NMC811 full batteries using DDE-PN. (e) Discharge profiles (0.1 C) of 500 mAh Li||NMC811 cells using DDE-PN electrolyte at different temperatures. (f) Comparison of a cell-level (output) energy density and power density with state-of-the-art electrolytes and our work at ultra-low temperature.